# Robust Target Association Method with Weighted Bipartite Graph Optimal Matching in Multi-Sensor Fusion

**DOI:** 10.3390/s26010049

**Published:** 2025-12-20

**Authors:** Hanbao Wu, Wei Chen, Weiming Chen

**Affiliations:** 1Faculty of Automation, Wuhan University of Technology, Wuhan 430070, China; 358923@whut.edu.cn (H.W.); 13863@whut.edu.cn (W.C.); 2Canglongdao Campus, Wuhan Digital Engineering Institute, Wuhan 430074, China; 3Faculty of Engineering, China University of Geosciences, Wuhan 430074, China

**Keywords:** group targets association, systematic error, cluster, machine learning, weighted bipartite graph, optimal matching

## Abstract

Accurate group target association is essential for multi-sensor multi-target tracking, particularly in heterogeneous radar systems where systematic biases, asynchronous observations, and dense formations frequently cause ambiguous or incorrect associations. Existing approaches often rely on strict spatial assumptions or pre-trained models, limiting their robustness when measurement distortions and sensor-specific deviations are present. To address these challenges, this work proposes a robust association framework that integrates deep feature embedding, density-adaptive clustering, and global graph-theoretic matching. The method first applies an autoencoder–HDBSCAN clustering scheme to extract stable latent representations and obtain adaptive group structures under nonlinear distortions and non-uniform target densities. A weighted bipartite graph is then constructed, and a global optimal matching strategy is employed to compensate for heterogeneous systematic errors while preserving inter-group structural consistency. A mutual-support verification mechanism further enhances robustness against random disturbances. Monte Carlo experiments show that the proposed method maintains over 90% association accuracy even in dense scenarios with a target spacing of 1.4 km. Under various systematic bias conditions, it outperforms representative baselines such as Deep Association and JPDA by more than 20%. These results demonstrate the method’s robustness, adaptability, and suitability for practical multi-radar applications. The framework is training-free and easily deployable, offering a reliable solution for group target association in real-world multi-sensor fusion systems.

## 1. Introduction

Group target association is a critical issue in real multi-sensor tracking systems, particularly in multi-radar surveillance, cooperative sensing networks, and congested airspace environments [1]. Targets frequently appear in formations or dense clusters, and these structures are essential for downstream inference tasks [2]. However, heterogeneous sensor biases, asynchronous observations, and the close proximity of targets can distort group geometry and create significant ambiguity in the measurement space [3]. Traditional approaches tend to break down under such conditions because they are sensitive to systematic errors, rely on assumptions of uniform density, and lack a unified mechanism that couples robust feature representation with global association constraints. As a result, association errors may propagate through the entire tracking pipeline. These challenges highlight the need for more resilient group-association strategies that remain accurate and consistent under complex sensing conditions.

Existing literature on group target association can be broadly categorized into statistical association algorithms and learning-based models. However, both categories face fundamental limitations when applied to heterogeneous multi-radar systems characterized by significant systematic errors. Traditional statistical methods, such as JPDA and MHI, are predominantly predicated on assumptions of spatial proximity [4,5,6] and motion continuity [7,8]. For example, ref. [9] proposed a filtering-based method to eliminate systematic errors and improve association accuracy. However, this method relies heavily on prior training, resulting in poor adaptability and limited performance in dynamic environments. Refs. [10,11] introduced a Random Matrix (RM)-based model that incorporates formation structure and the number of group targets to estimate the centroid and individual target positions, effectively addressing the positional offset caused by systematic errors. Nevertheless, as the group target density increases, the formation structure becomes less distinguishable, making it difficult to maintain high association accuracy, particularly in high-density environments. Refs. [12,13] proposed a multi-target tracking approach based on Random Finite Set (RFS) theory and Bayesian filters (BN), extending the Gaussian Mixture Probability Hypothesis Density (GM-PHD) algorithm to group target motion scenarios. This method improves association accuracy to some extent, but its neglect of accurate estimation for individual targets limits its applicability in complex environments. Refs. [14,15,16,17] leveraged the weak invariance of relative positional relationships under systematic error conditions and proposed methods such as point cloud alignment, formation matching, and position referencing to construct group target association models. These methods have shown promising results in sparse target scenarios, but their effectiveness degrades as target density increases due to instability in relative positional relationships. Refs. [18,19] treated systematic errors as translations and rotations of targets and employed techniques such as phase estimation and Fourier transforms for alignment and association. Ref. [20] proposed a method using commonly observed targets by sensors, incorporating fused centroid positions and headings as references to construct topological description vectors, and used gray relational analysis as a test statistic for robust global association decision-making. While effective in single-sensor scenarios with Gaussian noise, these methods degrade rapidly under systematic biases. Systematic errors introduce nonlinear offsets causing the entire group structure to shift in the measurement space. In such instances, the “nearest neighbor” criterion often leads to incorrect associations, as probabilistic filters lack the global constraints necessary to correct for rigid body displacements inherent in group biases.

Recent research on group target association has increasingly focused on data-driven approaches, including both deep learning and graph-based models. Several studies combine clustering with classical tracking to enhance association performance from different perspectives. Clustering analysis has been integrated with the Hough transform to construct multi-formation trajectory initialization schemes, yielding relatively stable tracking across diverse formation patterns [21]. This idea has been further developed by embedding an improved Hough transform into a density-based spatial clustering of applications with noise (DBSCAN)-based framework to reinforce trajectory initialization and improve robustness in dense target environments [22]. Other studies adopt split-and-merge tracking mechanisms on top of DBSCAN clustering to better handle closely spaced targets and reduce association errors under high-density conditions [23]. Nearest-neighbor matching has also been incorporated into graph-based formulations to refine trajectory initialization, improving tracking stability while reducing computational complexity [24]. In parallel, advances in deep learning have introduced graph neural networks (GNNs) and representation learning into the association process. Methods using GNNs with cross-attention mechanisms [25] or update-enhanced message passing [26] formulate association as an edge-classification problem, exploiting relational structures to improve cross-frame consistency. Bi-directional embedding approaches [27] further learn discriminative feature spaces to accommodate complex trajectory patterns. Although these methods achieve competitive performance in controlled or homogeneous sensing environments, they rely on learned statistical regularities and are therefore vulnerable to distribution shifts. In heterogeneous multi-radar systems, where each sensor may exhibit distinct and time-varying systematic biases, the geometric representations learned during training often fail to generalize without substantial retraining on scenario-specific data. Moreover, unlike optimization-based formulations that explicitly impose global consistency constraints, end-to-end learning models generally lack the flexibility to adapt to unseen measurement distortions at inference time. As a result, their performance can degrade considerably when the spatial structure of measurements departs from the conditions observed during training.

To address these limitations, this study proposes a robust association framework that explicitly decouples feature rectification from matching logic. Conventional linear dimensionality-reduction tools are inadequate for this purpose. Techniques such as PCA assume linear correlations and struggle to capture the nonlinear manifold distortions introduced by polar-to-Cartesian conversion errors and heterogeneous radar biases. For this reason, an autoencoder is employed to provide a nonlinear embedding; its activation functions allow the extraction of latent representations that remain stable across distorted sensor views. At the grouping stage, fixed-parameter clustering methods such as DBSCAN are prone to fragmentation or merging errors when target formations exhibit varying densities due to sensor resolution and bias effects. Hierarchical density-based spatial clustering of applications with noise (HDBSCAN) is therefore adopted to model cluster stability across multiple scales and to adaptively identify coherent groups in non-uniform density environments.

On this basis, we develop a graph-theoretic association method that integrates deep feature embedding with global optimal matching via a weighted bipartite graph. By enforcing global consistency constraints, the proposed method adaptively compensates for heterogeneous systematic errors and preserves group-level structure. The robustness of this framework is demonstrated through controlled simulations and evaluations on the nuScenes and KITTI benchmarks, where it achieves significant improvements over statistical baselines and state-of-the-art deep learning-based alternatives in complex sensing environments.

## 2. Target Grouping

In multi-sensor tracking systems, group target association relies on an accurate and reliable grouping stage that organizes individual measurements into coherent target groups. This grouping task becomes particularly challenging when sensors exhibit heterogeneous systematic errors or when targets move in dense formations. Under such conditions, the raw measurement space may become distorted, causing originally separable groups to overlap or lose structural clarity. An effective grouping mechanism is therefore essential, as it directly influences both the robustness of subsequent association and the overall computational load of the tracking pipeline.

To address these challenges, this study adopts a grouping strategy that integrates deep feature learning with density-adaptive clustering. The proposed Autoencoder–HDBSCAN framework is designed to extract stable latent representations from high-dimensional measurement data and to construct group structures without requiring a predefined number of clusters. The autoencoder component provides a nonlinear embedding that mitigates distortions caused by systematic biases and preserves the separability of different target groups, even when their measurements partially overlap. By compressing the data into a compact latent space, the autoencoder produces features that remain consistent across sensors and more suitable for clustering.

Building on these learned representations, HDBSCAN clustering is employed to identify group structures in a density-adaptive manner. Unlike conventional clustering methods that assume uniform density or require a fixed number of clusters, HDBSCAN can naturally accommodate varying target densities, detect noise points introduced by asynchronous sensing, and preserve meaningful group boundaries. This makes it particularly well suited for MS-MTT scenarios involving non-uniform formations and trajectory interactions. Together, the autoencoder and HDBSCAN form a cohesive grouping framework that improves robustness and supports reliable downstream group association under complex sensing conditions.

Specifically, the encoder compresses high-dimensional input data step by step into a low-dimensional latent space through a series of fully connected layers, aiming to eliminate noise and redundant information using low-dimensional representations, while extracting and retaining key information describing the targets. The encoder is composed of one input layer and four encoding layers:

**Input layer:** receives the original high-dimensional data;

**Encoding layers 1–3:** consist of 256, 128, and 64 neurons, respectively, using ReLU activation functions to achieve feature transformation, data compression, and deep feature extraction through nonlinear mappings;

**Encoding layer 4:** contains 32 neurons and adopts a linear activation function to output low-dimensional features.

The decoder uses a similar network structure to map the low-dimensional latent variables back to the original data space, achieving high-quality reconstruction of the input data. This provides feedback signals for the overall system and helps the encoder capture and retain more key information. The decoder is symmetrical to the encoder and consists of three decoding layers and one output layer:

**Decoding layers 1–3:** consist of 64, 128, and 256 neurons, respectively, with ReLU activation functions;

**Output layer:** uses a linear activation function to ensure that the dimensionality of the reconstructed data matches that of the original input.

The training of the encoder primarily relies on reconstruction loss, as defined in Equation (1). During training, the network learns effective representations by minimizing the discrepancy between the input data and the reconstructed data. Through this process, the network is able to adaptively capture and retain nonlinear features within the data under an unsupervised learning framework, enabling the autoencoder to discover complex patterns and structures in the data without requiring label information.(1)$$\begin{array}{cc} L= & L_{\mathrm{rec}}+L_{\mathrm{reg}}\\ L_{\mathrm{rec}}= & \frac{1}{N}\sum_{i=1}^{N}\left\|x_{i}-{\hat{x}}_{i}\right\|\\ L_{\mathrm{reg}}= & \lambda {\left\|\theta \right\|}^{2} \end{array}$$
where $L_{\mathrm{rec}}$ denotes the reconstruction loss, $L_{\mathrm{reg}}$ represents the regularization term, $x_{i}$ is the original input, ${\hat{x}}_{i}$ is the reconstructed output, N indicates the length of the input data, θ denotes the model parameters, and λ is the regularization coefficient.

HDBSCAN is a hierarchical density-based clustering algorithm, which serves as an extension and improvement of DBSCAN. This algorithm not only inherits DBSCAN’s strengths in discovering clusters of arbitrary shapes and handling noisy data, but also overcomes DBSCAN’s sensitivity to parameter selection (such as the fixed radius *ϵ*). The core idea of HDBSCAN is to construct a density hierarchy of the data to capture clustering patterns under different density levels and to automatically extract those clusters that remain stable throughout the hierarchy, while effectively distinguishing noise data. The detailed procedure is as follows:

**Step 1:** Density Representation. For each point in the dataset, the core distance is first calculated, which is defined as the distance from the point to its MinPts-th nearest neighbor, given a specified minimum number of points (MinPts). Meanwhile, the mutual reachability distance between any two points is defined as the maximum among the core distances of the two points and their actual Euclidean distance, as shown in Equation (2).(2)$$d_{\mathrm{mreach}-k}\;(x,y)=\max\{{\mathrm{core}}_{k}\;(x),{\mathrm{core}}_{k\;}(y),d(x,y)\}$$

HDBSCAN transforms the traditional Euclidean distance into mutual reachability distance, which incorporates local data density information into the original distance metric. This transformation reduces the influence of noise and outliers on distance measurement, allowing the algorithm to more accurately capture the unevenness in local data density.

**Step 2:** Construction of the Minimum Spanning Tree. Using mutual reachability distance as the edge weight, a complete graph is constructed among the data points. Then, the minimum spanning tree (MST) algorithm is applied to extract an acyclic graph that connects all data points. This step reveals the fundamental density-based connections between data points. The minimum spanning tree provides a global perspective of the connections among data points and lays the foundation for building the subsequent hierarchical tree.

**Step 3:** Hierarchical Clustering and Dendrogram Construction. Based on the minimum spanning tree, its edges are sorted in ascending order according to mutual reachability distance. By progressively increasing the distance threshold, edges are “cut” one by one, causing the graph to form separate connected components. These components naturally define the merging sequence under different thresholds. This merging process is recorded to form a dendrogram, where each node represents a cluster merged at a specific distance threshold, and the leaf nodes correspond to the original data points. In this process, clusters formed at different density levels are represented as distinct branches, reflecting the clustering structure of the data at multiple scales.

**Step 4:** Cluster Stability Measurement and Dendrogram Condensation. In the generated dendrogram, each cluster exhibits a different “lifespan” at various thresholds. By measuring the stability of a cluster—typically defined as the length of the distance interval over which it exists—its reliability can be quantified. Subsequently, based on a predefined minimum cluster size, small clusters that do not meet the requirement are pruned, and the hierarchical clustering tree is condensed. This process removes clusters with low stability and short lifespan, and labels isolated points as noise, resulting in a more concise hierarchical clustering structure. This process is also referred to as the construction of a condensed tree.

**Step 5:** Extraction of Final Clustering Results. Based on the stability metrics of each cluster in the dendrogram, the algorithm automatically selects the optimal clustering partition, ultimately dividing the data into multiple stable clusters. Data points that are not assigned to any stable cluster are labeled as noise.

This density-based hierarchical processing enables HDBSCAN to automatically determine the number of clusters, adapt to data distributions with varying densities, and exhibit strong robustness to noise.

## 3. Construction of Weighted Bipartite Graph for Targets

The essence of the group target association problem lies in establishing an optimal matching relationship between the pending group targets and the candidate group targets. Therefore, it can be modeled as a typical bipartite graph matching problem. In this context, one set of nodes represents the pending group targets, while the other set represents the candidate group targets. The weights of the edges in the graph represent the similarity or association cost between group targets.

To better reflect the multidimensional feature associations between group targets, the weighted bipartite graph constructed in this study comprehensively calculates edge weights based on multiple types of feature information, including spatial position features, velocity features, and target spacing multipliers. This approach enables a more comprehensive evaluation of the quality of matchings between group targets.

**Definition 1:** 
*If the vertex set *$E\left(G\right)$ *of a graph* G* can be divided into two non-empty subsets *$U_{1}$ *and* $U_{2}$*, and the edges between the vertices *$u_{i}$ *in* $U_{1}$ *and vertices* $v_{j}$ *in* $U_{2}$ *are assigned weights, then the graph *G* is a weighted bipartite graph* [28].

Based on the autoencoder-HDBSCAN clustering algorithm, spatial targets are divided into multiple target groups. Each target group can be easily split into a Pending Group Target set and a Candidate Group Target set using unique identifiers. As defined in Definition 1, let $E\left(G\right)$ denote the *k*-th target group, $U_{1}=\left\{u_{i}\right\}$ the Pending Group Target set, and $U_{2}=\left\{v_{j}\right\}$ the Candidate Group Target set. $u_{i}$, $v_{j}$ respectively represent targets in the pending and candidate sets. Accordingly, a target association bipartite graph G is constructed, where the vertices of the graph represent the targets in sets $U_{1}$ and $U_{2}$, and the edge weights reflect the degree of association between targets. The specific structure is illustrated in Figure 1.

To refine the construction of the weighted bipartite graph for target association, the feature vector x, y, z, $v_{x}$, $v_{y}$, $v_{z}$, $a_{x}$, $a_{y}$, $a_{z}$ of target *U*, *V* is represented as a new vertex in the bipartite graph, as $U_{1}=\left\{u_{i}\right\}=\left\{x_{i},y_{i},z_{i},v_{x_{i}},v_{y_{i}},v_{z_{i}},a_{x_{i}},a_{y_{i}},a_{z_{i}}\right\}$, $U_{2}=\left\{v_{j}\right\}=\left\{x_{j},y_{j},z_{j},v_{x_{j}},v_{y_{j}},v_{z_{j}},a_{x_{j}},a_{y_{j}},a_{z_{j}}\right\}$. Meanwhile, each edge in the bipartite graph is assigned a weight. $\omega =\left\{\lambda_{k}\right\}$, $\sum_{k=1}^{l}\lambda_{k}=1$ Where x, y, z, $v_{x}$, $v_{y}$, $v_{z}$, $a_{x}$, $a_{y}$, $a_{z}$ respectively represent the target’s distance, velocity, and acceleration along the x, y, and z axes, $i\leq N_{1}$, $j\leq N_{2}$, l=9. To simplify the computation, it is temporarily assumed that $N_{1}\leq N_{2}$.

When calculating the weighted value of vertices, the following conditions must be satisfied:

**Condition 1:** 
*Only vertices with the same dimensional features in both set *$U_{1}$ *and* $U_{2}$ *have non-zero weighted values; otherwise, the weight is zero. This implies that the association weight between targets is meaningful only when their features are similar.*

**Condition 2:** 
*At least one of the two sets, *$U_{1}$ *or* $U_{2}$*, must contain vertices with non-zero weighted values for all elements.*

**Definition 2:** 
*In the bipartite graph* 
*G, given sets *
$U_{1}$ *and*  $U_{2}$*, the weighted value* $C^{k}$ *between any two vertices* $u_{i}\in U_{1}$ *and* $v_{j}\in U_{2}$ *is calculated as:*(3)$$C^{k}=\lambda_{k}\cdot \frac{\underset{i,j}{\max}\left|v_{j}^{k}-u_{i}^{k}\right|-\left|v_{j}^{k}-u_{i}^{k}\right|}{\underset{i,j}{\max}\left|v_{j}^{k}-u_{i}^{k}\right|-\underset{i,j}{\min}\left|v_{j}^{k}-u_{i}^{k}\right|}$$

**Definition 3:** 

*The affinity between any two targets from the pending group target and the candidate group target is defined as:*



(4)
$$\prod \left(u_{i},v_{j}\right)=\sum_{k=1}^{l}C^{k}$$


Using Equation (4), the affinity $\prod \left(u_{i},v_{j}\right)$ between any two targets can be calculated and denoted as $\prod_{ij}$, forming the target affinity matrix, as shown in Figure 2.

Property 1: $0<\prod_{ij}\leq 1$.

The greater the affinity $\prod_{ij}$ between two targets $u_{i}$, $v_{j}$, the stronger the association between them; conversely, a smaller value indicates a weaker association. By analyzing and optimizing the affinity matrix, we can further improve the accuracy and efficiency of group target association, providing a theoretical foundation and algorithmic support for subsequent target matching and tracking.

## 4. Optimal Matching Association Method

System errors in sensor systems are a primary cause of confusion in group target association. Particularly under conditions of significant system error, the uncertainty in target association [29] increases substantially, greatly intensifying the difficulty of group target association. This issue becomes especially prominent in practical engineering scenarios, where phenomena such as multi-target fusion leading to false batches or mismatches frequently occur, and the impact of errors is even more severe.

To address this challenge, this study proposes a method based on the concept of optimal matching in weighted bipartite graphs. By leveraging a globally optimal matching strategy, the method effectively mitigates the influence of system errors, thereby improving the accuracy of group target association, as illustrated in Figure 3.

**Definition 4:** 
*Given the intimacy *$\prod_{ij}\geq 0$* between any two targets *$u_{i}$*, *$v_{j}$*, the objective of the intimacy matrix is to find a complete matching scheme that maximizes the sum of all target intimacies, which is referred to as the optimal matching* [20].

According to Definition 4, the group target association problem can be formulated as the following optimization problem:(5)$$\begin{array}{c} \max\sum_{i=1}^{N_{1}}\sum_{j=1}^{N_{2}}\prod_{ij}B_{ij},\\ s.t.\sum_{i}B_{im}=1,m=1,2,\cdots ,N_{2}\\ \sum_{i}B_{nj}=1,n=1,2,\cdots ,N_{1} \end{array}$$
where $B_{ij}=$ 1 or 0, $B_{ij}=1$ indicates that the two targets are in an optimal matching relationship, and $B_{ij}=0$ indicates that they are not.

According to Equation (5), the optimal matching problem of the bipartite graph can be transformed into a single-objective optimal linear programming problem. By solving for the optimal solution, the best target association can be achieved. To solve this optimization problem, this study uses the improved KM algorithm (Hungarian Algorithm), which has a computational complexity of $O\left(N_{2}^{3}\right)$ ensuring both solution accuracy and computational efficiency. The specific steps of the algorithm are as follows:

**Step 1:** Perform matrix augmentation by expanding the $N_{1}\times N_{2}$ dimensional intimacy matrix to an $N_{2}\times N_{2}$ dimensional matrix. When $N_{1}\leq i\leq N_{2}$, let $\prod \left(i,j\right)=0$, where the target set of the rows is $V\left(m\right)$ and the target set of the columns is $W\left(n\right)$.

**Step 2:** Initialize the feasible labels. $match\left(n\right)$ represents the matching relationship between sets $V\left(m\right)$ and $W\left(n\right)$, where the value of $H\left(match\left(n\right),n\right)$ is 1; The initial value of $match\left(i\right)$ is set to −1. Scan the rows and columns of the intimacy matrix to calculate the maximum value $\prod \left(i,j\right)$ in each row. Then, initialize the feasible labels $l_{x}\left(i\right)=\max\prod \left(i,j\right)$ and $ly\left(j\right)=0$ for the set V and W.

**Step 3:** Define the termination condition. Scan $V\left(m\right)$ and initialize the scan markers $sx\left(i\right)=false$, $sy\left(i\right)=false$. If $m\geq N_{2}$ the algorithm ends and outputs the result $H\left(match\left(i\right),i\right)$. Otherwise, proceed to Step 4. If a match is found, update accordingly m=m+1. If no match is found, proceed to Step 5.

**Step 4:** Find an augmenting path. Let $sx\left(m\right)=true$ and scan $W\left(n\right)$. If $sy\left(n\right)=false$ and $l_{x\left(m\right)}+l_{y\left(n\right)}=\prod \left(u,v\right)$, then update $sy\left(n\right)=true$. If $match\left(n\right)=-1$ or find an augmenting path from $V\left(match\left(n\right)\right)$, then update $match\left(n\right)=m$ and find the widening track from $V\left(m\right)$. Otherwise, the widening track is not found from $V\left(m\right)$.

**Step 5:** Matching Relationship Search. Initialize temporary variable *del*, and scan the rows and columns of the matrix. If $sx\left(i\right)=false$ and $sy\left(j\right)=false$, then update $del=\min\left(l_{x\left(i\right)}+l_{y\left(j\right)}-\prod \left(i,j\right),del\right)$. Scan each row *i*, if $sx\left(i\right)=true$, then update the feasible label $l_{x\left(i\right)}=l_{x\left(i\right)}+del$. Scan each column *j*, if $sy\left(j\right)=true$, then update the feasible label $l_{y\left(j\right)}=l_{y\left(j\right)}-del$. Then return to Step 3.

## 5. Verification of Target Association

Based on the above method, the optimal matching relationships among targets can be obtained from the intimacy matrix:(6)$$\begin{array}{l} F\left(v_{i},w_{j}\right)=\left\{\left(i,j\right)\right\}\\ \;i=1,2,\cdots ,N_{1}\\ \;j=1,2,\cdots ,N_{2}\\ \;B_{ij}=1 \end{array}$$

In multi-target tracking, system errors are one of the key factors affecting the accuracy of target association. Especially under large system errors, even if an apparently optimal matching relationship $F\left(v_{i},w_{j}\right)$ is obtained through the algorithm, it is difficult to set a unified decision threshold to accurately determine whether two radar-detected targets belong to the same batch. This is due to the uncertainty of the bias introduced by system errors, which makes a single threshold unsuitable for adapting to error disturbances in various scenarios.

Through the analysis of a large amount of real detection data from multi-source sensors, it has been found that there exists a certain mutual support relationship in the intimacy between targets in real association scenarios. Specifically, if a target has a high intimacy with its matched target, then the neighboring matching pairs often show a similar level of intimacy.

Based on this statistical feature, this paper introduces an intimacy consistency verification method based on threshold clustering to further improve the reliability of target association. This method determines whether the matching results exhibit association consistency according to the clustering results of intimacy values. In other words, if the intimacy values of most matching pairs in the optimal matching relationship are concentrated within a specific clustering range, the target association is considered valid; otherwise, it is regarded as non-associated. See Figure 4 for illustration.

The specific clustering algorithm proceeds as follows:

**Step 1:** Set an initial clustering discrimination threshold ε. Randomly select an intimacy value a from the target intimacy matrix as the seed value to form a cluster $e_{1}$. For each unclustered intimacy value b outside $e_{1}$, perform similarity evaluation. If $\left|b-a\right|\leq \varepsilon$, mark b as a non-seed value and include it in cluster $e_{1}$; otherwise, proceed to Step 2.

**Step 2:** Select a new seed value from the remaining unclustered intimacy values to form a new cluster $e_{2}$, and repeat Step 1 until all intimacy values are either included in a cluster or no longer satisfy the clustering condition.

**Step 3:** If any cluster $e_{*}$ still contains non-seed values that have not been fully evaluated, repeat Steps 1 and 2 until no new clusters can be generated. The algorithm then terminates.

After clustering, if the number of intimacy values in a single cluster accounts for more than a predefined discrimination threshold *θ* of all matched pairs, the current optimal matching result is considered to have high consistency. Consequently, a reliable association between target groups is determined. Otherwise, it is assumed that the deviation caused by system errors is substantial, the current matching lacks sufficient association confidence, and thus the target groups are considered unassociated.

The proposed method leverages the statistical consistency of target intimacy to effectively correct for system-induced deviations, thereby improving association accuracy and enhancing robustness in complex environments. 

In summary, the overall flowchart of the proposed algorithm is illustrated as Figure 5.

## 6. Simulation Verification and Analysis

### 6.1. Simulation Environment

In this study, a simulation scenario is constructed involving three spatially distributed 2D radars, located at coordinates ( −30 km, 7 km, 0 km), (0 km, 10 km, 0 km), and (40 km, 6 km, 0 km), respectively. The detection errors of the radars are assumed to follow Gaussian distributions $N\left(0,\sigma^{2}\right)$, with the measurement accuracies for range and azimuth given as follows: (50 m, 0.1°), (80 m, 0.15°), and (100 m, 0.2°), respectively. All radars operate with a uniform sampling interval of 2 s, and their startup times are staggered with a 0.5 s delay between each radar. All sensors observe a shared airspace region.

To evaluate the proposed method, we generate a synthetic dataset consisting of four target groups with a total of 17 airborne tracks. One group contains dispersed targets, while the remaining three maintain structured formations. The simulation spans 150 time cycles. Although the baseline model assumes subsonic motion, the actual trajectories used in the experiments are not restricted to uniform linear movement. As illustrated in Figure 6, several groups exhibit nonlinear patterns such as curved paths, turning maneuvers, and trajectory intersections. Some groups also display partial overlaps and varying motion headings, introducing complex interactions commonly encountered in realistic multi-sensor multi-target tracking scenarios. The initial spacing between adjacent targets is set to 1.4 km. This simulation setting is intentionally designed to reflect the multi-radar sensing conditions considered in this work, in which heterogeneous systematic biases, asynchronous measurements, and non-collinear sensor placements play a central role in shaping group-association behavior.

The performance of the proposed association algorithm is evaluated using the target association accuracy (TAA) metric [30], computed as the average over *L* Monte Carlo trials:(7)$$\rho_{\mathrm{TTA}}=\frac{\sum_{i=1}^{L}\frac{N}{M}}{L}\times 100\%$$

Here, N denotes the number of fused target points that are closest in spatial distance to the true targets, and M represents the total number of fused target points. The target association accuracy (TAA) primarily reflects the overall similarity between the fused targets and the true targets, serving as an indicator of the association algorithm’s effectiveness in correctly aligning measurements with actual targets.

To further validate the robustness of the proposed framework beyond synthetic environments and to demonstrate its applicability to real-world sensing conditions, we extended our evaluation to include two widely recognized public benchmarks: the nuScenes dataset and the KITTI tracking benchmark. While the simulation scenarios described above are essential for isolating and quantifying specific systematic errors, they inherently simplify the noise landscape. Real-world data allows us to challenge the algorithm with non-Gaussian noise, heavy environmental clutter, and intricate object interactions that are difficult to model artificially. This complementary testing ensures the proposed framework remains effective not just under controlled biases, but also when facing the genuine irregularities of physical sensing systems.

### 6.2. Performance Comparison and Analysis

A total of 50 Monte Carlo simulation runs were conducted in this experiment. The weight parameters were set as $\lambda =\left\{0.20,0.20,0.20,0.10,0.10,0.10,0.03,0.03,0.03\right\}$, and the clustering threshold was set to ε=0.15. The proposed method was compared against JPDA, MHI and Deep Association under various target scenarios. The specific experimental setup is described below.

**Scenario 1:** Target Environment with Systematic Error in the Same Direction. In this scenario, systematic errors in the same direction were superimposed on the radar measurements, causing target detections from different radars to be biased toward the same direction. Specifically, Radars 1 and 2 were subjected to systematic errors in range and azimuth described by the functions $\Delta d1\left(n\right)=25n$, $\Delta \alpha_{1}\left(n\right)=0.1n$, respectively. Radar 3 had systematic errors described by $\Delta d2\left(n\right)=50n$, $\Delta \alpha_{2}\left(n\right)=0.2n$, where n=1,2,3,⋯,10. A total of 10 flight missions were simulated under this condition, and the results are presented as Figure 7.

**Scenario 2:** Target Environment with opposite direction systematic error. In this scenario, opposite-direction systematic errors were superimposed on the radar measurements, causing the detected targets from different radars to be displaced in opposite directions. Specifically, Radar 1 had systematic errors described by the functions $\Delta d1\left(n\right)=-25n$, $\Delta \alpha_{1}\left(n\right)=0.1n$ in range and azimuth, respectively. Radars 2 and 3 had systematic errors described by the functions $\Delta d2\left(n\right)=50n$, $\Delta \alpha_{2}\left(n\right)=0.2n$ in range and azimuth, respectively. Where n=1,2,3,⋯,10. A total of 10 flight missions were simulated under this condition, and the results are presented as Figure 8.

**Scenario 3:** Algorithm Robustness Verification with Varying Target Flight Separation. In this scenario, the robustness of the algorithm is validated by varying the flight separation between targets. The target flight separation is modeled by the function $\Delta d\left(n\right)=1400-100n$. Where n=1,2,3,⋯,10. A total of 10 flight missions were simulated under this condition, and the results are presented as Figure 9.

Comprehensive analysis of experimental results:

a. From the experimental data in Figure 7 and Figure 8, it is observed that as the radar system errors increase proportionally, the radar detection targets begin to show displacement. Specifically, when the same-direction system error reaches n=3, the radar targets exhibit significant displacement. Similarly, a displacement phenomenon is observed when the reverse system error reaches n=1. The traditional JPDA, fails to effectively compensate for the target position offsets caused by system errors, resulting in a rapid decrease in association accuracy as system errors increase. In contrast, MHI maintains a target association accuracy of over 70% under system error conditions. This is primarily due to the uncertainty in estimating target rotation and translation caused by sensor random errors, leading to approximately 20% of targets being misassociated. As system errors intensify, the geometric shapes of targets undergo further distortion, causing the association accuracy to decrease. Notably, Deep Association and the global optimal matching method based on weighted bipartite graphs proposed in this paper, consistently maintain a target association accuracy of over 90%. This indicates that both methods exhibit high robustness in resisting system error interference, effectively achieving correct associations among multiple targets.

b. As shown in the experimental results in Figure 9, with the continuous decrease in target flight separation, the influence of sensor random errors becomes more pronounced, amplifying the uncertainty in target association. When the amplitude of random errors approaches the actual separation distance between targets, the JPDA’s association accuracy plummets from over 90% to around 20%. This is due to the fact that in small separation conditions, random errors obscure the true motion trajectory of the targets, leading to a significant number of incorrect associations. The denser the target distribution, the higher the error rate. Although the MHI still maintains an association accuracy above 70% in this environment, it shows a slight decline compared to other experimental conditions. Deep Association also experiences a decrease in association accuracy to around 70% due to the accumulation of errors from previous steps. In contrast, the global optimal matching method based on target bipartite graphs proposed in this paper effectively decouples and globally optimizes the target frames, making it more robust to random error interference in densely clustered environments. As a result, the association accuracy remains stable at over 90%, demonstrating high universality and reliability.

c. From the experimental data in Figure 9, it can be observed that when the target overlap increases and mutual interference intensifies, the performance of all methods diverges significantly. Specifically, traditional local matching methods, such as JPDA, struggle to correctly identify the true motion trajectory of targets in the presence of overlaps and occlusions, leading to a drastic decrease in association accuracy. In contrast, the strategies proposed in MHT and Deep Association exhibit strong robustness in handling global information but still show a decline in accuracy under intense target interference. By integrating global perspectives and mutual support relations between targets, the global optimal matching method based on weighted bipartite graphs proposed in this paper significantly reduces the risk of misassociation due to target overlap. This method maintains an association accuracy of over 90% even in high-interference environments, further validating the superiority and stability of the global optimal matching strategy in complex multi-target scenarios.

d. In dynamic target scenarios, the rapid changes in target motion states, including speed fluctuations and trajectory crossings, become significant factors influencing association accuracy. When the target motion trajectory intersects or undergoes abrupt changes, traditional local matching methods fail to capture the motion trend in time, resulting in a sharp drop in association accuracy. Although the MHI offers some resistance to random motion error interference, it still has limitations. In contrast, the method proposed in this paper, through effective decoupling between consecutive frames and incorporating global optimal matching, accurately captures the target motion trend. This effectively reduces the impact of accumulated motion errors. The experimental results show that even under conditions of drastic changes in target motion states, the method still maintains a high association accuracy, demonstrating its practical and stable performance in dynamic multi-target tracking applications.

In conclusion, from the comprehensive comparison and analysis of the 30 Monte Carlo simulation experiments conducted across three experimental environments, it is evident that the target association method based on the global optimal matching of weighted bipartite graphs significantly outperforms traditional methods in resisting sensor measurement errors. This method not only provides reliable target association results under various system and random error conditions but also exhibits superior performance in complex target environments. These findings underscore the high engineering application potential of this approach.

To substantiate this engineering potential and bridge the gap between theoretical models and physical deployment, we extended our evaluation to include the nuScenes and KITTI benchmarks.

Table 1 summarizes the performance on public benchmarks and shows a consistent trend with our simulation findings. Across both datasets, the proposed method attains the highest association accuracy, demonstrating strong generalization beyond the controlled simulation environment. In the nuScenes benchmark, where raw radar measurements contain substantial clutter and sparsity, our approach (0.924) surpasses the Deep Association baseline (0.887), largely due to the Autoencoder–HDBSCAN grouping module’s ability to suppress noise and restore coherent group structures. Traditional methods such as JPDA perform considerably worse (0.732), as their local probabilistic models tend to diverge under the heavy uncertainty characteristic of millimeter-wave radar. On the KITTI dataset, although the performance gap between methods narrows due to the cleaner and more structured nature of the trajectory data, the proposed method still achieves the best accuracy (0.936). This result highlights the method’s robustness in preserving global structural consistency, even in scenarios that do not exhibit the systematic error patterns emphasized in our simulation studies.

### 6.3. Algorithm Complexity Analysis

To compare the computational performance of the proposed method with three other methods, a simulation experiment was conducted to evaluate the time complexity of the four methods. The experiment was performed by repeating a single loop 1 million times, with the average runtime of all loops taken to minimize random errors. The tests were conducted under the following hardware and software environment: an Intel Core i7–10,700 processor with a clock speed of 2.9 GHz, 8 cores, 16 threads, and 12 MB L3 cache, ensuring high parallel computing capabilities; 16 GB DDR4 memory with a frequency of 3200 MHz; a 512 GB solid-state drive (SSD), significantly accelerating data read and write speeds; and an NVIDIA GeForce GTX 4070 discrete graphics card to assist with accelerating certain computational tasks. In terms of the software environment, the operating system was 64-bit Windows 10 Professional, with Visual Studio 2019 as the development environment. Algorithm implementation and data analysis were carried out on the Python 3.8 platform, utilizing common data processing and visualization libraries such as NumPy, SciPy, and Matplotlib. This detailed configuration ensured the stability and efficiency of the experimental environment, providing a solid hardware and software foundation for testing and validating complex algorithms.

Figure 10 shows the average CPU runtime of the four methods under different target quantities. The experimental results indicate that as the number of targets increases, the runtime for all methods shows a noticeable upward trend. Among them, Deep Association and the one proposed in this paper had relatively high computation times, mainly due to the global optimization strategies they employed, resulting in a worst-case time complexity of $O\left(n^{3}\right)$. In contrast, MHI and JPDA utilized only local neighborhood information, leading to a lower computational complexity, typically around $O\left(n^{2}\right)$. Although global optimization methods incur higher computational costs, statistical analysis of the extensive simulation results shows that all four methods can meet the practical engineering requirements for the target scales tested. The difference in actual runtime reflects the tradeoff between global optimality and local efficiency in algorithm design and provides a reference for selecting the appropriate target association method for different scenarios.

### 6.4. Ablation Experiments

To rigorously evaluate the contribution of each component and justify the methodological choices, a comprehensive set of ablation experiments was conducted under ten levels of systematic error. The proposed framework was compared against widely used baselines for feature extraction (PCA, UMAP) and clustering (DBSCAN, Spectral Clustering). Eight model variants were evaluated to isolate the effects of feature embedding, clustering strategy, and association verification. Table 2 summarizes the comparative results.

The findings show that the Autoencoder-based embedding consistently outperforms PCA and UMAP across all error conditions. PCA retains moderate stability at lower error levels but experiences clear degradation as spatial distortions increase. UMAP, which relies on manifold preservation, exhibits pronounced sensitivity to global sensor-induced distortions, with accuracy dropping to 0.785 under high-error multipliers. In contrast, the Autoencoder achieves the most stable and accurate performance (up to 0.993), demonstrating its superior ability to learn nonlinear representations that remain robust under heterogeneous and non-uniform distortions.

The clustering module plays a critical role when target density varies due to systematic bias. Replacing HDBSCAN with DBSCAN results in the largest accuracy drop, particularly when error levels cause uneven spatial stretching; DBSCAN’s fixed neighborhood parameters fail to adapt to such changes. Spectral Clustering performs moderately but still lags behind the proposed design. HDBSCAN delivers the most reliable performance by automatically adjusting to density variations and effectively identifying group structures even under severe measurement distortions.

A separate comparison examines the role of the association verification step. Removing this module yields consistent accuracy degradation across all error levels. Without verification, mismatches accumulate especially in scenarios involving overlapping trajectories, as the model cannot filter inconsistent affinity relationships. Incorporating the verification mechanism significantly improves stability, confirming its necessity in preventing erroneous pairings from propagating through the association process.

Overall, the ablation results demonstrate that the strength of the proposed framework lies in the complementary contributions of its components. The Autoencoder offers a discriminative and distortion-resilient feature space superior to PCA and UMAP; HDBSCAN provides density-adaptive grouping that surpasses DBSCAN and Spectral Clustering; and the weighted bipartite graph with verification ensures global structural consistency. While each module contributes uniquely, their integrated operation yields a system that is substantially more robust and accurate than any individual variant or baseline approach.

## 7. Conclusions

(1) This paper presents a robust target grouping framework that integrates autoencoder-based deep feature embedding with density-adaptive HDBSCAN clustering. The autoencoder provides a nonlinear low-dimensional representation that mitigates distortions caused by heterogeneous systematic errors, while HDBSCAN identifies group structures without requiring a predefined number of clusters. Compared with traditional clustering approaches, the proposed method offers superior noise tolerance, density adaptiveness, and cluster separability, thereby establishing a reliable foundation for subsequent group target association.

(2) Building on this grouping stage, we propose a weighted bipartite graph association model that incorporates an affinity matrix and a global optimal matching formulation. By enforcing global consistency constraints and compensating for systematic deviations across sensors, the method effectively resolves association ambiguities that arise in dense formations and overlapping measurements. Experimental evaluations demonstrate that the proposed approach consistently outperforms representative baselines, achieving more than a 20% improvement over Deep Association in dense-target scenarios.

(3) To enhance robustness under random disturbances, a mutual-support verification mechanism is introduced to adaptively refine affinity-based clustering and suppress spurious associations. Simulation results show that even when inter-target spacing approaches the level of sensor measurement noise, the proposed method maintains an association accuracy of 92.6%, representing an improvement of approximately 70 percentage points over JPDA, and confirming the stability of the model under challenging sensing conditions.

(4) The proposed framework does not rely on pre-trained models or predefined motion assumptions, making it highly suitable for heterogeneous multi-radar environments where large labeled datasets are unavailable and systematic biases vary across sensors. Future work will focus on improving computational efficiency for real-time applications, extending the method to fully asynchronous observation settings, and validating the approach on additional engineering datasets involving complex sensor geometries.

## Figures and Tables

**Figure 1 sensors-26-00049-f001:**
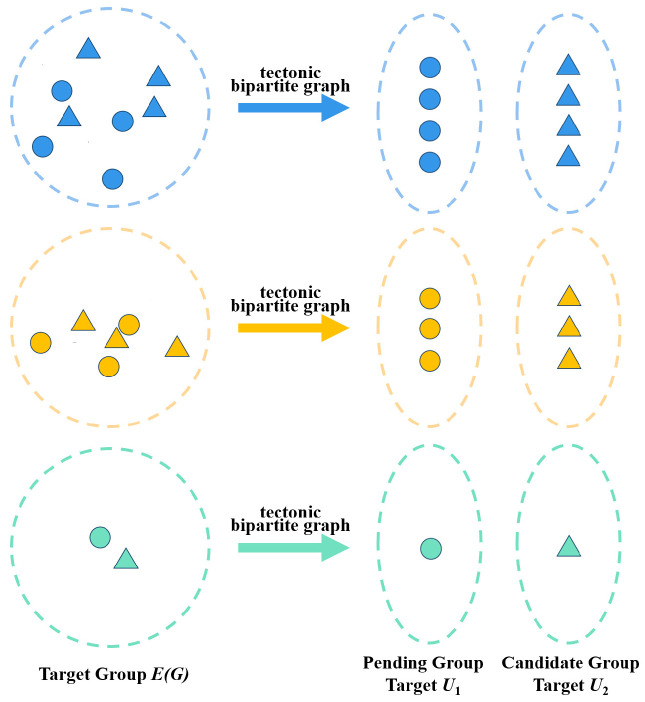
Schematic diagram of target association bipartite graph construction.

**Figure 2 sensors-26-00049-f002:**
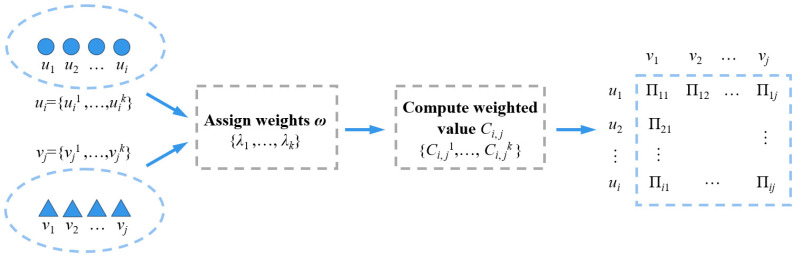
Schematic diagram of intimacy matrix for group targets.

**Figure 3 sensors-26-00049-f003:**
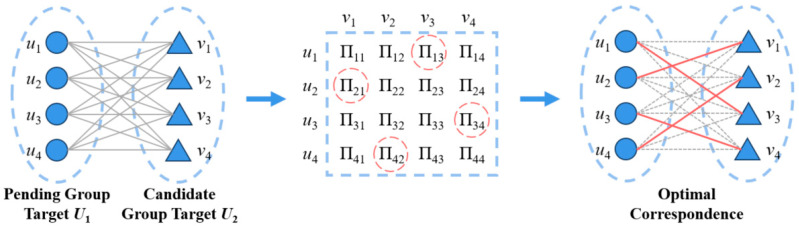
Schematic diagram of target association by optimal matching.

**Figure 4 sensors-26-00049-f004:**
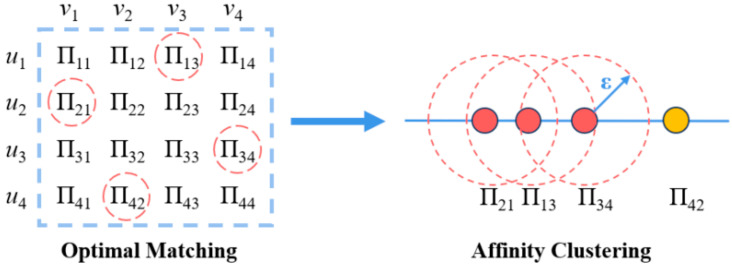
Verification for target association by optimal matching.

**Figure 5 sensors-26-00049-f005:**
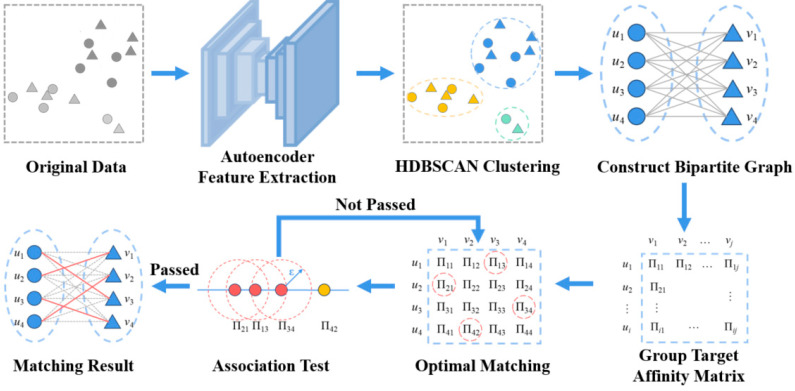
The progress of target association algorithm.

**Figure 6 sensors-26-00049-f006:**
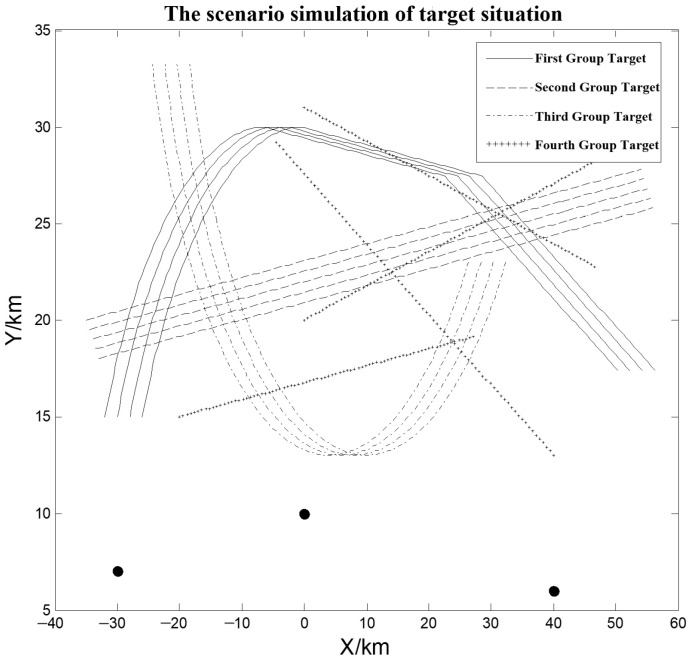
The scenario simulation of target situation.

**Figure 7 sensors-26-00049-f007:**
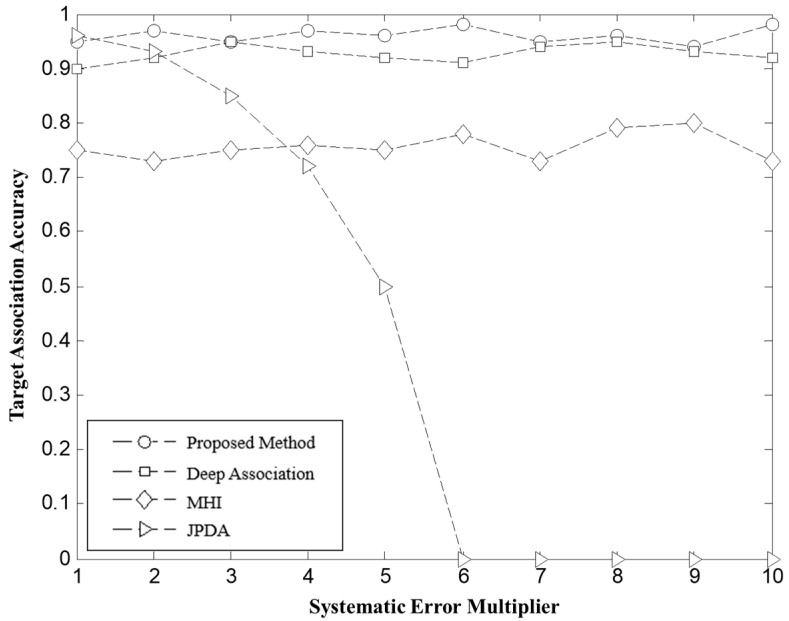
Target association accuracy at different unidirectional systematic error ratio.

**Figure 8 sensors-26-00049-f008:**
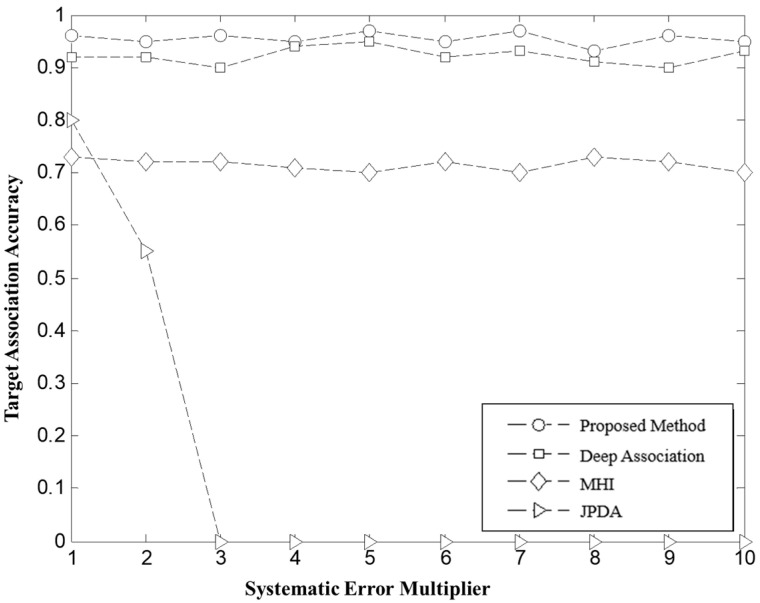
Target association accuracy at different reverse systematic error ratio.

**Figure 9 sensors-26-00049-f009:**
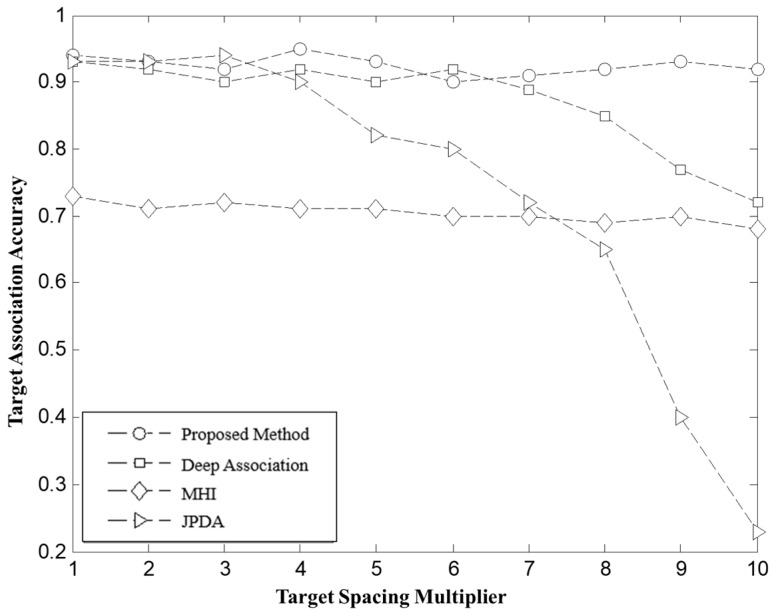
Target Association Accuracy at Different Spacing Ratios.

**Figure 10 sensors-26-00049-f010:**
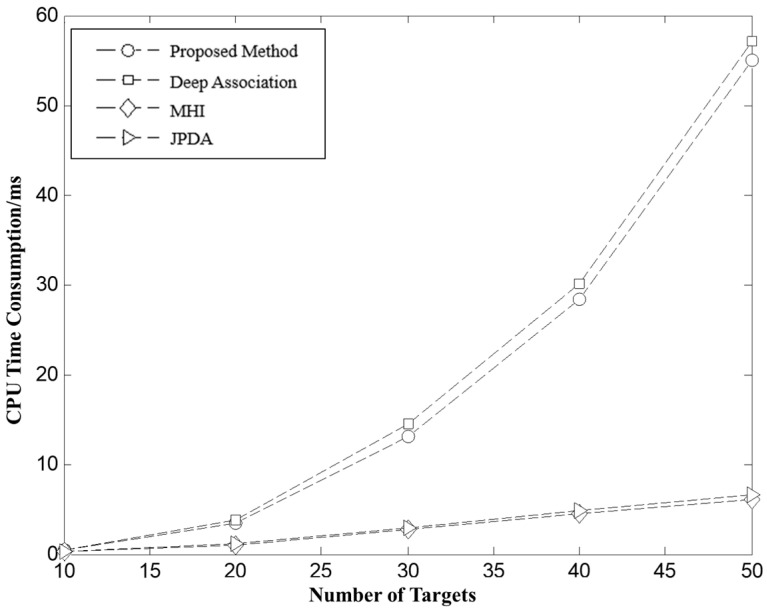
The complexity comparison of different algorithm.

**Table 1 sensors-26-00049-t001:** Comparison of target association accuracy on real-world datasets.

Method	nuScenes	KITTI
Proposed Method	0.924	0.936
Deep Association	0.887	0.901
MHI	0.815	0.843
JPDA	0.732	0.795

**Table 2 sensors-26-00049-t002:** Results of ablation experiment.

Model	Systematic Error Multiplier									
	1	2	3	4	5	6	7	8	9	10
Proposed Method	0.942	0.957	0.939	0.966	0.951	0.992	0.94	0.943	0.937	0.993
PCA + HDBSCAN + Weighted Bipartite Graph + association test	0.935	0.948	0.931	0.952	0.940	0.965	0.915	0.910	0.895	0.935
UMAP + HDBSCAN + Weighted Bipartite Graph + association test	0.912	0.925	0.898	0.910	0.885	0.890	0.820	0.785	0.760	0.785
Autoencoder + Spectral Clustering + Weighted Bipartite Graph + Association Test	0.928	0.935	0.915	0.930	0.910	0.925	0.855	0.840	0.825	0.865
Autoencoder + DBSCAN + Weighted Bipartite Graph + Association Test	0.915	0.928	0.895	0.905	0.875	0.880	0.750	0.735	0.710	0.755
Autoencoder + HDBSCAN + Weighted Bipartite Graph	0.931	0.945	0.924	0.946	0.921	0.942	0.860	0.853	0.837	0.883

## Data Availability

Data will be made available on request.

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
