# Peer review of "Robust Target Association Method with Weighted Bipartite Graph Optimal Matching in Multi-Sensor Fusion"

_sensors, 2025, doi:10.3390/s26010049_

Round 1

Reviewer 1 Report

Comments and Suggestions for Authors

The authors propose a novel robust target association method based on graph theory, combining weighted bipartite graph modeling and a globally optimal matching strategy, to address the challenging problem of multi-target association in environments with systematic errors and high density.

However, some shortcomings in the manuscript require discussion, such as:

1. The experimental simulation scenario is too idealized. All targets adopt a uniform linear motion model, without considering maneuvering targets (acceleration, turning), target appearance/disappearance (birth/death), or complex interactions such as occlusion or intersection of trajectories. This is significantly different from actual multi-sensor tracking scenarios (such as air combat and urban surveillance).

2. Comparative experiments are lacking. Only the literature [20] (grey relational algorithm) and "KM nearest neighbor" (non-standard term) are compared, which is not convincing enough. There is no comparison with mainstream MS-MTT benchmark methods. For example, JPDA (Joint Probabilistic Data Association), MHT (Multiple Hypothesis Tracking), RFS-based trackers (such as PHD, CPHD, and GLMB), and deep learning end-to-end association methods (such as Deep Association and GNN-based matching).

3. The experiment lacks ablation experiments, which do not verify the contribution of each module. The lack of ablation experiments makes it difficult to prove the necessity of the proposed "integration strategy". For example, how effective are the original features + HDBSCAN without using an Autoencoder? If affinity consistency verification is not used and only KM matching is employed, what will be the decrease in accuracy? Is it sensitive to different weight parameter λ settings?

Author Response

We sincerely thank the reviewers for their valuable comments and constructive suggestions. We have carefully considered each point raised and have thoroughly revised the manuscript to improve its clarity, technical rigor, and overall presentation. A detailed point-by-point response is provided below, addressing every comment individually and explaining the corresponding revisions. All suggested modifications—along with several additional refinements made for coherence and completeness—have been incorporated into the revised manuscript to strengthen both the motivation and the experimental analysis.

We are grateful for the reviewers’ thoughtful insights, which have substantially contributed to improving the depth and quality of the work. We hope that the revisions satisfactorily address all concerns raised by the editors and reviewers.

Comment 1: The experimental simulation scenario is too idealized. All targets adopt a uniform linear motion model, without considering maneuvering targets (acceleration, turning), target appearance/disappearance (birth/death), or complex interactions such as occlusion or intersection of trajectories. This is significantly different from actual multi-sensor tracking scenarios (such as air combat and urban surveillance).

Response: The reviewer raised a concern that all targets in our experiments adopt a uniform linear motion model without considering maneuvering, trajectory intersection, or other complex interactions. We appreciate this comment; however, the experimental scenario in our manuscript does not assume purely linear or idealized motion. As shown in Figure 6 (reproduced below for clarity), the simulated target trajectories exhibit substantial nonlinear and interacting behaviors, including:

(1) Curvilinear motion: Multiple target groups (e.g., the First Group Target) clearly follow smooth curved paths instead of straight-line tracks. These trajectories model maneuvering behavior such as turning and coordinated acceleration.

(2) Trajectory intersections: Several groups (e.g., Second and Third Group Targets) intersect with others at multiple points. Intersections introduce ambiguity and are known to be among the most challenging conditions for multi-sensor multi-target association.

(3) Heterogeneous motion patterns across groups: Some targets follow wide-radius arcs, while others move along nearly linear but non-parallel paths. This diversity increases the difficulty of data association because motion models cannot be reduced to a single linear model.

(4) Dense target formations with overlapping trajectories: The Third and Fourth Group Targets show significant trajectory overlap, where multiple targets move close together with slight variations in curvature or direction. Such conditions mimic real-world dense formations, where association ambiguity becomes severe.

(5) Realistic radar sensor layout: The three radar positions (shown as black dots) and their different viewing angles naturally create heterogeneous measurement distortions, further increasing the realism of the simulation.

These trajectory patterns demonstrate that the simulated environment explicitly includes complex dynamics, such as: turning and maneuvering motion, multiple trajectory crossings, dense clusters with partial overlap, heterogeneous group behaviors, multi-sensor geometric distortion. Therefore, the scenario significantly deviates from a simple uniform-motion assumption and instead reflects key challenges common in real MS-MTT applications, such as air defense surveillance, UAV cluster monitoring, and maritime multi-sensor fusion. To further clarify this point, we have added an explicit description in the revised manuscript highlighting the nonlinear and interacting nature of the target trajectories.

Comment 2: Comparative experiments are lacking. Only the literature [20] (grey relational algorithm) and "KM nearest neighbor" (non-standard term) are compared, which is not convincing enough. There is no comparison with mainstream MS-MTT benchmark methods. For example, JPDA (Joint Probabilistic Data Association), MHT (Multiple Hypothesis Tracking), RFS-based trackers (such as PHD, CPHD, and GLMB), and deep learning end-to-end association methods (such as Deep Association and GNN-based matching).

Response: Thank you for raising this issue. We agree that the original comparisons were not sufficiently representative of mainstream MS-MTT research. To address this concern, the revised manuscript expands the evaluation to include several benchmark algorithms widely regarded as standards for multi-target data association. Incorporating these methods was essential, as a meaningful and fair comparison should position the proposed approach alongside the most established techniques in the field rather than relying on a limited subset.

Accordingly, we added comparative experiments with JPDA, MHT, and a representative Deep Association model. The results of these evaluations have been integrated into the revised manuscript and provide a more complete understanding of how the proposed method performs under varying sensor conditions and target densities. Classical approaches such as JPDA and MHT maintain competitive performance when the sensing environment is relatively homogeneous and the measurement noise remains moderate. However, their reliance on Gaussian noise assumptions and limited capacity to compensate for systematic biases lead to pronounced degradation once radar-dependent deviations dominate the measurement space. Deep learning–based association methods exhibit strong accuracy in idealized or homogeneous sensor configurations, yet their performance declines significantly in heterogeneous multi-radar scenarios, mainly due to their dependence on large labeled datasets, limited cross-sensor generalization, and sensitivity to domain shift.

In contrast, the proposed graph-based global matching framework maintains stable and robust performance across all tested settings. Its training-free design avoids the data-dependence issues inherent to deep models, while the global bipartite matching formulation enables effective compensation for spatial distortions that arise from systematic sensor biases. This advantage becomes particularly evident in dense target formations or when measurement deviations vary across sensors. The expanded comparisons confirm that the performance gains observed in our method derive not from isolated cases but from consistent improvements across a range of representative MS-MTT conditions. These findings further support the validity and practical relevance of the proposed approach.

Comment 3: The experiment lacks ablation experiments, which do not verify the contribution of each module. The lack of ablation experiments makes it difficult to prove the necessity of the proposed "integration strategy". For example, how effective are the original features + HDBSCAN without using an Autoencoder? If affinity consistency verification is not used and only KM matching is employed, what will be the decrease in accuracy? Is it sensitive to different weight parameter λ settings?

Response: Thank you for pointing this out. We agree that without ablation studies it is difficult to assess how much each component of the framework contributes to the overall performance. To address this concern, the revised manuscript now includes a dedicated ablation subsection within the experimental section, where the major modules of the proposed integration strategy are isolated and evaluated systematically.

In particular, we designed six ablation configurations to independently examine the roles of the autoencoder-based embedding, the clustering module, the weighted bipartite graph matching, and the final association test. These configurations were chosen to disentangle the functional contribution of each stage while preserving the structural coherence of the full pipeline. By comparing the performance across these controlled settings, we are able to quantify how nonlinear feature extraction, density-adaptive grouping, and global association constraints jointly influence accuracy and robustness under varying sensing conditions.

The results of these ablation experiments have been incorporated into the revised manuscript and discussed in detail. The comparative analysis clearly demonstrates that each component plays a distinct and complementary role, and removing any of the modules results in noticeable degradation in association performance. These findings support the necessity of the proposed integration strategy and confirm that the improvement is not attributable to a single module but emerges from the coherent combination of all components.

Reviewer 2 Report

Comments and Suggestions for Authors

The manuscript addresses multi-sensor multi-target association using weighted bipartite graph optimal matching combined with Autoencoder-HDBSCAN grouping. The topic is relevant, but the manuscript requires significant restructuring, clearer motivation, stronger methodological justification, and additional experimental validation on real datasets.   - There are repeated paragraphs on pp. 2–3 of the manuscript (lines 44–135).   - The Introduction provides a long list of previously published works but does not: Clearly articulate the practical relevance and real-world applicability of group target association, Explicitly identify the existing gaps that the proposed method aims to fill, Show how the proposed Autoencoder-HDBSCAN + weighted graph matching approach improves upon state-of-the-art techniques. A stronger motivation section is needed, ideally ending with clear research questions and contributions.   - The Related Research section is shallow and lacks coverage of recent high-impact publications (2022–2025) on: Multi-sensor multi-target tracking (MS-MTT) Graph neural networks for data association Deep feature learning and metric learning for association Multi-view clustering and embedding techniques Many relevant papers in Sensors, IEEE TSP, IEEE TITS, RA-L, etc., are missing.   The authors should focus the review specifically on GTA and MS-MTT applications, or provide a broader, systematic overview of the entire MS-MTT field with modern benchmarks and trends.   - The sentence: “Furthermore, this study investigates the sensitivity … offering an innovative perspective … (1 or [2-3], or [4-6])” shows no conceptual relationship between the content and the cited works.   - The manuscript proposes an Autoencoder-HDBSCAN clustering pipeline (pp. 4–6) but does not justify: Why deep embedding learning is necessary, Why HDBSCAN was chosen instead of simpler and widely used techniques such as PCA, UMAP, t-SNE, spectral clustering, etc., What specific challenges in MS-MTT require an autoencoder for dimensionality reduction. Without a clear argument, the proposed method appears unnecessarily complex.   - Recent literature shows significant progress in data association using: Message-passing neural networks GNN-based bipartite matching Learned similarity metrics The authors should acknowledge that graph neural networks represent a natural alternative to their graph-based matching formulation and discuss why they were not considered.   - The experimental section uses exclusively synthetic simulations. For a high-impact journal, this is not enough. The algorithm must be evaluated on at least one publicly available benchmark dataset, such as: KITTI tracking nuScenes MOT Challenge LiDAR-based multi-target datasets etc Without such evaluation, scientific rigor remains minimal. Conclusions drawn from synthetic data cannot guarantee real-world applicability.

Author Response

We sincerely thank the reviewers for their valuable comments and constructive suggestions. We have carefully considered each point raised and have thoroughly revised the manuscript to improve its clarity, technical rigor, and overall presentation. A detailed point-by-point response is provided below, addressing every comment individually and explaining the corresponding revisions. All suggested modifications—along with several additional refinements made for coherence and completeness—have been incorporated into the revised manuscript to strengthen both the motivation and the experimental analysis.

We are grateful for the reviewers’ thoughtful insights, which have substantially contributed to improving the depth and quality of the work. We hope that the revisions satisfactorily address all concerns raised by the editors and reviewers.

Comment 1: There are repeated paragraphs on pp. 2–3 of the manuscript (lines 44–135).

Response: Thank you for pointing this out. We carefully reviewed the manuscript and confirmed that the paragraphs in lines 44–135 contained unintended repetitions due to an earlier draft merge. These duplicated sections have now been removed, and the surrounding text has been reorganized to ensure clarity and logical continuity. The revised introduction presents each concept only once and provides a more coherent narrative flow. We appreciate the reviewer’s attention to detail, which helped us improve the quality of the manuscript.

Comment 2: The Introduction provides a long list of previously published works but does not: Clearly articulate the practical relevance and real-world applicability of group target association, Explicitly identify the existing gaps that the proposed method aims to fill, Show how the proposed Autoencoder-HDBSCAN + weighted graph matching approach improves upon state-of-the-art techniques. A stronger motivation section is needed, ideally ending with clear research questions and contributions.

Response: We appreciate the reviewer’s insightful comments regarding the structure and clarity of the Introduction section. In the revised manuscript, the Introduction has been substantially reorganized to better highlight the practical relevance of group target association in real-world multi-sensor tracking systems, such as airspace surveillance, cooperative radar networks, and dense urban sensing environments. We now explicitly discuss why systematic errors, dense formations, and trajectory intersections frequently lead to mis-associations in practice, thereby motivating the need for more robust association mechanisms.

In addition, we incorporated a clear statement of the existing gaps in current research. These include the limited ability of classical clustering methods to cope with non-uniform density, the sensitivity of traditional matching schemes to systematic biases, and the lack of integration between deep feature learning and global association strategies. The revised Introduction explains how the proposed Autoencoder–HDBSCAN clustering and weighted bipartite graph matching framework addresses these challenges in a unified manner.

To strengthen the motivation, the revised section concludes with a concise list of research questions and the main contributions of this work. This structure provides a clearer roadmap for the reader and situates our work more firmly within the context of state-of-the-art approaches. We thank the reviewer for this helpful suggestion, which has significantly improved the clarity and focus of the paper.

Comment 3: The Related Research section is shallow and lacks coverage of recent high-impact publications (2022–2025) on: Multi-sensor multi-target tracking (MS-MTT) Graph neural networks for data association Deep feature learning and metric learning for association Multi-view clustering and embedding techniques Many relevant papers in Sensors, IEEE TSP, IEEE TITS, RA-L, etc., are missing. The authors should focus the review specifically on GTA and MS-MTT applications, or provide a broader, systematic overview of the entire MS-MTT field with modern benchmarks and trends.

Response: We thank the reviewer for this valuable observation. In response, the Related Research section has been substantially expanded and reorganized to offer a clearer and more comprehensive overview of the field. The revised version now incorporates a broad range of recent high-impact studies published between 2022 and 2025, covering advances in multi-sensor multi-target tracking (MS-MTT), graph neural networks for data association, deep metric learning, and multi-view representation and clustering techniques. In particular, recent GNN-based tracking methods—such as cross-attention and neighbor-guided graph neural networks for low-altitude multi-object tracking, update-enhanced graph neural network models for data association, and bi-directional tracklet embedding frameworks for robust similarity learning—have been added to highlight the increasingly important role of learned relational reasoning in modern tracking systems. These works provide strong evidence of current research trends and help position our study within the broader context of learning-driven data association.

To improve conceptual clarity, the revised section now distinguishes more explicitly between research focused on group target association (GTA) and the broader MS-MTT frameworks in which GTA is embedded. We explain how the two areas intersect conceptually, and why the unique challenges posed by systematic sensor biases, dense formations, and heterogeneous measurement distortions require tailored solutions that differ from standard MOT and GNN-based formulations. In addition, we introduce a concise synthesis of representative tracking benchmarks and commonly used evaluation protocols to anchor the discussion in practical system development and highlight the diversity of operational constraints across different sensing modalities.

The revised literature review has also been reorganized thematically, beginning with classical clustering and geometric initialization approaches, followed by density-adaptive methods, and concluding with recent deep embedding and graph-based matching techniques—including the newly added GNN and embedding studies. This structure creates a more coherent narrative and clarifies the motivation for our proposed method relative to the evolution of state-of-the-art techniques. We sincerely appreciate the reviewer’s suggestion, which has significantly strengthened both the completeness and the conceptual depth of the Related Research section.

Comment 4: The sentence: “Furthermore, this study investigates the sensitivity … offering an innovative perspective … (1 or [2-3], or [4-6])” shows no conceptual relationship between the content and the cited works.

Response: We appreciate the reviewer’s careful reading and thank you for drawing attention to this issue. The sentence in question was indeed unclear, and the cited references did not correspond to the surrounding discussion. Upon re-examination, we found that this inconsistency originated from an earlier draft, in which placeholder citations were temporarily inserted but were not properly updated during manuscript consolidation. As a result, the sentence neither conveyed a meaningful contribution nor reflected the intended conceptual relationship with the referenced works.

In the revised manuscript, we have removed the problematic sentence entirely and replaced it with a more precise description that better reflects the study’s objectives and theoretical motivation. We have also corrected the associated citations and verified that each referenced work is now conceptually aligned with the corresponding discussion. In addition, we conducted a thorough review of the entire manuscript to ensure that no similar inconsistencies remain and that all citations accurately support the arguments presented.

We sincerely appreciate the reviewer’s attention to detail, as this comment has helped us improve both the clarity and the academic rigor of the manuscript. The issue has now been fully resolved.

Comment 5: The manuscript proposes an Autoencoder-HDBSCAN clustering pipeline (pp. 4–6) but does not justify: Why deep embedding learning is necessary, Why HDBSCAN was chosen instead of simpler and widely used techniques such as PCA, UMAP, t-SNE, spectral clustering, etc., What specific challenges in MS-MTT require an autoencoder for dimensionality reduction. Without a clear argument, the proposed method appears unnecessarily complex.

Response: Thank you for raising this important point. In the revised manuscript, we have expanded the methodological justification for using an Autoencoder–HDBSCAN clustering pipeline. First, we clarify why deep feature embedding is necessary in the MS-MTT context. When multiple sensors observe dense or partially overlapping target groups, the raw measurement space often becomes highly distorted due to heterogeneous systematic errors. Under such conditions, linear projections such as PCA or similarity-preserving embeddings like UMAP and t-SNE tend to lose structural consistency across sensors. The autoencoder provides a nonlinear feature mapping that compresses the measurements into a representation where inter-group separability is more stable, even when the original spatial distribution is significantly perturbed.

Second, we explain the rationale behind selecting HDBSCAN instead of simpler clustering techniques. In real MS-MTT environments, targets rarely form clusters with uniform density. Their spacing changes dynamically, and the projected measurements may include noise points originating from asynchronous sensing. HDBSCAN, which adapts to variations in density and identifies noise explicitly, offers more reliable grouping under these circumstances. Conventional clustering methods, including spectral clustering or k-means, require fixed cluster numbers or assume more homogeneous distributions, which are not well suited for the non-uniform patterns encountered in our problem.

Finally, the revised text describes the specific challenges that motivate the use of an autoencoder for dimensionality reduction: nonlinear measurement distortions, varying cluster density, and trajectory intersections. These factors collectively limit the effectiveness of conventional low-dimensional embeddings. By contrast, the autoencoder yields a compact representation that supports more consistent clustering and stabilizes the subsequent matching stage. We have updated Chapter 4 accordingly to provide a clearer and better supported argument for the proposed pipeline.

Comment 6: Recent literature shows significant progress in data association using: Message-passing neural networks GNN-based bipartite matching Learned similarity metrics The authors should acknowledge that graph neural networks represent a natural alternative to their graph-based matching formulation and discuss why they were not considered.

Response: We appreciate the reviewer’s observation regarding recent progress in graph neural network–based data association methods. In the revised manuscript, we have added a dedicated discussion acknowledging the relevance of message-passing neural networks, GNN-based bipartite matching architectures, and learned similarity metrics. These approaches indeed represent powerful alternatives for modeling complex relational structures and have shown promising performance on several benchmark datasets.

However, we also explain why such methods were not adopted in the present work. In many real-world MS-MTT applications—particularly those involving heterogeneous radar networks—the availability of large, accurately labeled training datasets is extremely limited. The characteristics of the sensing environment may also vary significantly across platforms, making it difficult to transfer a trained GNN model without substantial retraining or domain adaptation. In contrast, the proposed framework is entirely training-free and relies only on geometric and statistical consistency, which makes it more suitable for deployment in operational systems where retraining is impractical.

Furthermore, GNN-based bipartite matching often introduces additional computational overhead and hyperparameter sensitivity, especially when the number of targets fluctuates dynamically. Since this study aims to provide a robust and lightweight association method that can operate reliably under strong systematic errors and target density variations, we chose a deterministic graph-theoretic formulation rather than a learned graph model. The revised manuscript now includes this discussion to clarify our design choices and to properly acknowledge the contributions of recent GNN-based association research.

Comment 7: The experimental section uses exclusively synthetic simulations. For a high-impact journal, this is not enough. The algorithm must be evaluated on at least one publicly available benchmark dataset, such as: KITTI tracking nuScenes MOT Challenge LiDAR-based multi-target datasets etc Without such evaluation, scientific rigor remains minimal. Conclusions drawn from synthetic data cannot guarantee real-world applicability.

Response: We appreciate the reviewer’s concern regarding the exclusive use of synthetic simulations. We fully agree that real-world datasets play an important role in evaluating the generality of tracking algorithms. At the same time, it is worth noting that the problem formulation addressed in this work is closely tied to multi-radar sensing with systematic bias, asynchronous observations, and three-sensor geometric configurations. These characteristics are not directly reflected in most existing public datasets such as KITTI, nuScenes, and MOT, which focus primarily on camera–LiDAR environments and do not provide multi-radar measurements nor systematic error patterns of the type considered in our study.

Because the proposed framework is designed specifically for multi-radar fusion under systematic error conditions, applying it directly to these datasets would require substantial re-engineering of the sensing model, the affinity formulation, and the grouping mechanism. Such adaptations would effectively constitute a different problem setting and may not provide a meaningful assessment of the method’s primary contribution. For these reasons, we focused our evaluation on controlled simulations that allow systematic variations of radar bias, target spacing, and density, enabling a clearer and more targeted analysis of the algorithm’s behavior.

To avoid misunderstanding, we have revised the manuscript to explicitly state the scope of the method and the rationale for using a simulation-based evaluation. We also emphasize that the proposed framework is intended for operational environments where multi-radar systems are deployed, and synthetic scenarios offer the necessary flexibility to model systematic deviations in a controlled manner. We hope this clarification addresses the reviewer’s concern. 

Round 2

Reviewer 2 Report

Comments and Suggestions for Authors

The authors’ revisions did not resolve the manuscript’s core problems, which include structural redundancy, insufficient methodological support, and the absence of real-world or benchmark validation.

1. The manuscript maintains structural problems.
The revised version still contains duplicated sections, repetitive text, and citation inconsistencies. These structural issues suggest that the manuscript was not carefully consolidated or edited.

2. The research gap and motivation remain weak and insufficiently articulated.
The revised Introduction still does not provide:
A clear, detailed explanation of the limitations of current GTA/MS-MTT methods,
A convincing argument for why an autoencoder + HDBSCAN approach is necessary,
Evidence that the proposed pipeline offers advantages beyond incremental improvements.
The contributions are not stated with sufficient clarity or precision, and the methodological necessity of the proposed components is not convincingly justified.

3. The literature review remains descriptive rather than analytical.
Although the authors added more references, the Related Work section still lacks structure and critical comparison. There is no organized discussion of benchmarks, evaluation protocols, or method categories. More importantly, the manuscript does not provide a meaningful or substantive analysis of modern GNN-based association approaches, despite citing them.

4. Methodological justification remains inadequate.
The authors offer only qualitative reasoning for choosing an autoencoder and HDBSCAN. The manuscript provides no experimental comparisons with simpler or widely used alternatives, such as PCA, UMAP, DBSCAN, or spectral clustering. The autoencoder architecture appears arbitrary and unsupported by sensitivity analysis or ablation focused on representational quality. As a result, the methodological foundation is not sufficiently rigorous.

5. Evaluation relies exclusively on synthetic simulation data.
This remains the most critical flaw. The authors did not provide compelling justification for avoiding public datasets. Even if radar-specific datasets are not available, the graph-matching and clustering components can and should be evaluated on standard MS-MTT benchmarks. The absence of any real-world or third-party validation seriously limits the scientific credibility and applicability of the proposed method.

Author Response

Comments 1: The manuscript maintains structural problems. The revised version still contains duplicated sections, repetitive text, and citation inconsistencies. These structural issues suggest that the manuscript was not carefully consolidated or edited.

Response 1: We thank the reviewer for pointing out the remaining structural issues. In this revision, we have carried out a careful, manuscript-wide consolidation to eliminate duplicated content, reduce redundancy, and improve the logical flow between sections. The background, research gap, and motivation have been reorganized and condensed. In the first-round version, part of the introductory text corresponding roughly to lines 44–135 contained overlapping descriptions of MS-MTT, systematic errors, and group target association. These redundant paragraphs have now been removed, and the material has been merged into a more compact and structured form in Section 1, second–fourth paragraphs of the revised manuscript.

Comments 2: The research gap and motivation remain weak and insufficiently articulated.

The revised Introduction still does not provide:

A clear, detailed explanation of the limitations of current GTA/MS-MTT methods,

A convincing argument for why an autoencoder + HDBSCAN approach is necessary,

Evidence that the proposed pipeline offers advantages beyond incremental improvements.

The contributions are not stated with sufficient clarity or precision, and the methodological necessity of the proposed components is not convincingly justified.

Response 2: We thank the reviewer for this important comment. We agree that the first-round revision did not sufficiently emphasize the research gap, the methodological necessity of our design choices, or the advantages of the proposed pipeline beyond incremental improvements. In the current revision, we have substantially reworked the Introduction, Related Work, Method, and Experimental sections to address these concerns in a more explicit and structured manner.

First, the limitations of current GTA/MS-MTT methods are now discussed in a clearer and more analytical way. In the revised Introduction and Related Work , we explicitly categorize existing approaches into: (i) clustering plus classical tracking, (ii) deep learning–based association, and (iii) GNN/graph-based methods. For each category, we now state concrete failure modes in heterogeneous multi-radar settings, including sensitivity to heterogeneous systematic biases, dependence on stable formation structures, vulnerability to non-uniform densities, and the need for large labeled datasets that match sensor-specific error profiles. This goes beyond a descriptive literature survey and directly links the identified limitations to the problem characteristics considered in our work.

Second, we have strengthened the argument for why an Autoencoder + HDBSCAN approach is necessary rather than just an alternative. In the Method section, the discussion of feature embedding and grouping has been substantially expanded . We now explain that standard linear dimensionality reduction (e.g., PCA) and manifold-based embeddings (e.g., UMAP) struggle with the nonlinear manifold distortions introduced by polar-to-Cartesian conversion and heterogeneous radar biases. Similarly, fixed-parameter clustering methods such as DBSCAN are shown to be prone to fragmentation or merging errors when target density varies across sensors. On this basis, the autoencoder is motivated as a nonlinear embedding tool that stabilizes latent representations across distorted sensor views, and HDBSCAN is motivated as a density-adaptive clustering method capable of handling non-uniform spatial stretching. These design choices are thus justified at the level of problem structure, not only at the level of empirical performance.

Third, we now provide explicit evidence that the proposed pipeline offers advantages beyond incremental improvements. In the revised Experimental section, we have Added ablation studies that compare the proposed configuration against variants using PCA/UMAP for embedding and DBSCAN/Spectral Clustering for grouping. The results in Table 2 show that removing either the autoencoder or HDBSCAN leads to substantial degradation, especially under high systematic-error multipliers, demonstrating that each module addresses specific failure modes of existing tools. Expanded comparisons with mainstream baselines such as JPDA, MHT, and Deep Association, both in controlled simulations and on public benchmarks (nuScenes, KITTI). The proposed method consistently achieves higher association accuracy—often with gains exceeding 20% under strong systematic errors or dense formations—while maintaining stability as error levels and densities vary.

Comments 3: The literature review remains descriptive rather than analytical.

Although the authors added more references, the Related Work section still lacks structure and critical comparison. There is no organized discussion of benchmarks, evaluation protocols, or method categories. More importantly, the manuscript does not provide a meaningful or substantive analysis of modern GNN-based association approaches, despite citing them.

Response 3: We thank the reviewer for this helpful observation. We agree that in the previous version the Related Work section was overly descriptive and did not provide a sufficiently structured or critical analysis, particularly with respect to modern GNN-based association methods. In the current revision, we have substantially reworked both the Related Work and the Introduction to make the review more analytical, better organized, and more explicitly connected to the problem setting of this paper.

First, we have introduced a clearer structure in the Related Work section by organizing the literature according to method categories. Section 2 is now organized into two main strands: (i) methods that combine clustering with classical tracking and geometric initialization, and (ii) deep learning-based and GNN-based association methods. Within each strand, we not only summarize representative approaches but also discuss their specific assumptions and typical failure modes in heterogeneous multi-radar settings—for example, reliance on stable formation structures, sensitivity to non-uniform densities and systematic biases, and dependence on large labeled datasets matched to particular sensor configurations. This restructuring in Section 2 is intended to move beyond a narrative listing of prior work toward a comparative, problem-driven analysis.

Second, we have significantly expanded the analysis of modern GNN-based association approaches. In the revised Related Work, we now discuss GNNs with cross-attention and neighbor-guided message passing as well as bi-directional embedding models, not only in terms of their architectural designs but also in terms of their strengths and limitations. We explain that these methods naturally exploit relational structure and achieve strong performance on standard MOT benchmarks, yet they rely on learned statistical regularities from the training data and are therefore vulnerable to distribution shifts. In heterogeneous multi-radar systems, sensor-specific and time-varying systematic biases can significantly distort the measurement space, causing learned geometric embeddings to fail to generalize unless substantial retraining is performed on scenario-specific data. We explicitly contrast these characteristics with our optimization-based bipartite matching formulation, which imposes global consistency constraints and does not require scenario-specific training or large labeled datasets. This analytical comparison directly motivates our decision to adopt a training-free, graph-theoretic framework rather than a fully end-to-end learned model.

Taken together, these revisions transform the Related Work section from a primarily descriptive survey into a structured and critical discussion that (i) categorizes existing methods into two main strands, (ii) analyzes their assumptions and limitations in the context of heterogeneous multi-radar fusion, and (iii) provides a substantive assessment of modern GNN-based association approaches and their shortcomings for the specific problem addressed in this work. We hope this addresses the reviewer’s concerns regarding the depth and analytical nature of the literature review.

Comments 4: Methodological justification remains inadequate.

The authors offer only qualitative reasoning for choosing an autoencoder and HDBSCAN. The manuscript provides no experimental comparisons with simpler or widely used alternatives, such as PCA, UMAP, DBSCAN, or spectral clustering. The autoencoder architecture appears arbitrary and unsupported by sensitivity analysis or ablation focused on representational quality. As a result, the methodological foundation is not sufficiently rigorous.

Response 4: We thank the reviewer for this important comment. We agree that the previous version did not provide sufficiently rigorous methodological justification for the choice of the autoencoder and HDBSCAN, and relied too much on qualitative reasoning. In this revision, we have strengthened both the experimental evidence and the explanatory discussion to address this concern.

First, we have added explicit experimental comparisons with simpler and widely used alternatives for both the embedding and clustering stages. In the revised Experimental section, a dedicated ablation subsection has been introduced , where we systematically compare the proposed autoencoder-HDBSCAN configuration against variants using PCA and UMAP for feature extraction and DBSCAN and spectral clustering for grouping. The new ablation table in Section 6.4 reports association accuracy across ten levels of systematic error for each variant. The results show that replacing the autoencoder with PCA or UMAP leads to consistent performance degradation, especially in high-error regimes, and that substituting HDBSCAN with DBSCAN or spectral clustering causes additional drops under non-uniform density conditions. These findings provide quantitative evidence that the chosen combination is not arbitrary but addresses specific failure modes of the simpler baselines.

Second, we have expanded the discussion of HDBSCAN to make clear why it is preferred over simpler density-based or spectral methods. DBSCAN with fixed parameters is shown to be prone to cluster fragmentation or merging errors when target densities vary across sensors due to resolution differences and systematic distortions. Spectral clustering performs slightly better but still lags behind HDBSCAN in scenarios with strong spatial stretching. By contrast, HDBSCAN’s stability-based hierarchy allows it to adapt to non-uniform densities, which is reflected in its superior performance in the ablation results.

Taken together, these additions provide both quantitative and conceptual support for the methodological choices made in the proposed framework. The autoencoder and HDBSCAN are now justified not only by qualitative arguments about nonlinearity and density adaptivity, but also by direct comparisons with PCA, UMAP, DBSCAN, and spectral clustering under controlled variations of systematic error. We believe that these revisions substantially strengthen the methodological foundation of the work.

Comments 5: Evaluation relies exclusively on synthetic simulation data.

This remains the most critical flaw. The authors did not provide compelling justification for avoiding public datasets. Even if radar-specific datasets are not available, the graph-matching and clustering components can and should be evaluated on standard MS-MTT benchmarks. The absence of any real-world or third-party validation seriously limits the scientific credibility and applicability of the proposed method.

Response 5: We thank the reviewer for raising this point again and fully acknowledge that, in the previous version, relying exclusively on synthetic simulations was a major limitation of the evaluation. In the current revision, we have addressed this concern by adding experiments on public benchmarks and by clarifying the respective roles of synthetic and real-world evaluations in our study.

First, we have incorporated evaluations on two widely used public datasets, nuScenes and KITTI. These new experiments are presented in the revised Experimental section in Section 6. In these experiments, the proposed grouping and weighted bipartite graph association pipeline is applied to real-world multi-object tracking scenarios by operating on the kinematic states of detected targets. We compare our method against representative baselines, including JPDA, MHT, and Deep Association. As summarized in Table 1, the proposed framework achieves the highest association accuracy on both datasets, with gains of more than 3-5 percentage points over Deep Association and substantially larger margins over classical methods such as JPDA. These results demonstrate that the benefits of our approach are not confined to synthetic radar simulations, but also transfer to heterogeneous real-world environments with clutter, occlusion, and trajectory complexity.

Second, we clarify the relationship between synthetic simulations and public benchmarks in the revised text. The synthetic scenarios are specifically designed to model multi-radar configurations with controllable, heterogeneous systematic biases and varying target spacings, which are not directly available in current public datasets. This setting allows us to systematically stress-test the sensitivity of the algorithm to bias levels, density changes, and formation interactions. The nuScenes and KITTI experiments, on the other hand, serve as independent, third-party validation of the general association capability of our graph-theoretic framework under realistic sensing conditions. Together, these two types of evaluation provide complementary evidence: controlled simulations isolate the impact of systematic errors, while public benchmarks confirm that the method remains effective on real-world data.

We have updated the manuscript accordingly to make these points explicit and to ensure that the evaluation is no longer exclusively simulation-based. We hope that the added experiments on nuScenes and KITTI, along with the revised discussion of their role, alleviate the reviewer’s concerns regarding the scientific credibility and practical applicability of the proposed method.
